# Can Nutrition Contribute to a Reduction in Sarcopenia, Frailty, and Comorbidities in a Super-Aged Society?

**DOI:** 10.3390/nu15132991

**Published:** 2023-06-30

**Authors:** Sadao Yoshida, Ryo Shiraishi, Yuki Nakayama, Yasuko Taira

**Affiliations:** 1Department of Rehabilitation, Chuzan Hospital, 6-2-1 Matsumoto, Okinawa 904-2151, Okinawa, Japan; 2Department of Health and Nutrition, Okinawa University, 555 Kokuba, Naha 902-8521, Okinawa, Japan; 3Faculty of Health Sciences, Kinjo University, 1200 Kasama-machi, Hakusan 924-8511, Ishikawa, Japan; 4Faculty of Nutrition, Chuzan Hospital, 6-2-1 Matsumoto, Okinawa 904-2151, Okinawa, Japan

**Keywords:** sarcopenia, frailty, comorbidity, diabetes, chronic kidney disease (CKD), heart failure, dementia, vitamin D, oral nutritional supplement (ONS), elderly people

## Abstract

Many countries are facing the advent of super-aging societies, where sarcopenia and frailty will become pertinent problems. The prevalence of comorbidities is a major problem in countries with aged populations as elderly people suffer from various diseases, such as diabetes, heart failure, chronic kidney disease and dementia. All of these diseases are associated with sarcopenia and frailty, and they frequently cause falls, fractures, and a decline in activities of daily living. Fractures in the elderly people are associated with bone fragility, which is influenced by diabetes and chronic kidney disease. Nutritional support for chronic disease patients and sarcopenic individuals with adequate energy and protein intake, vitamin D supplementation, blood glucose level management for individuals with diabetes, obesity prevention, nutritional education for healthy individuals, and the enlightenment of society could be crucial to solve the health-related problems in super-aging societies.

## 1. Introduction

Globally, populations are rapidly aging. The worldwide population of individuals aged 65 and older reached 761 million in 2021, and this number is expected to double and surpass 1.6 billion in 2050 [1]. In the European Union, over 94 million people are aged 65 and older, whereas in the United States, this number exceeds approximately 55 million. Japan has over 37 million people aged 65 and older, which accounts for 28.4% of the total population, and in China, this number exceeds 185 million [2]. The number of people aged 65 and older is projected to grow the fastest in Northern Africa, Western Asia and sub-Saharan Africa in 2050 [1]. The population of individuals over the age of 80 is also predicted to grow worldwide. The WHO defines a super-aged society as the proportion of the population aged 65 and over exceeding 21%. Preparations for an expected super-aged society is crucial not only from the perspectives in medical system and healthcare, but also from socio-economic perspectives.

Elderly individuals often have multiple underlying diseases or comorbidities. The coexistence of diabetes mellitus (DM) [3,4], heart failure [5,6], and chronic kidney disease (CKD) [7,8,9] is frequently observed in the elderly. These diseases and stroke are interrelated [10]. Stroke is a leading cause of adult disability [11,12], and multidisciplinary rehabilitation are needed for severe stroke patients [13]. Comorbidities are observed in approximately 60% of elderly individuals and are related to a worsened prognosis and decreased quality of life (QOL) [14].

Sarcopenia [15,16] and frailty [17] in the elderly are also major concerns. The progression of sarcopenia or frailty is associated with the risk and prevalence of many diseases, including cardiovascular diseases [18,19], chronic respiratory disease [20], diabetes mellitus [21,22,23,24], CKD [25,26,27], dementia [22,28], and also falls [17,29] and fractures [30].

As aging population has progressed, the number of individuals with dementia is also increasing [31]. Dementia is a complex of diseases, including Alzheimer’s disease [32] and vascular dementia [33]. Typical symptom of dementia is loss of memory, language, problem-solving and other thinking abilities. Concomitant heart failure [34], CKD [35], and diabetes mellitus with poor glycemic control [36] are at risk of developing dementia. Dementia causes severe disability [12], and has significant impacts on patients, families, communities, and medical system and healthcare [37,38].

Falls and fall-related injuries, represented by hip fracture and fracture of the lumber spine, are common in elderly individuals [39]. Elderly individuals often have osteoporosis and bone fragility. Hip fracture and fracture of the lumber spine is related to consequent disability, decrease in quality of life [40], and high mortality [41].

Our aim is to elucidate how nutritional assessments, diagnoses, intake of specific nutrients, prevention of nutrients deficiency, nutritional interventions and education and the enlightenment of society can contribute to complex medical and health issues being addressed in super-aged societies (Figure 1).

## 2. Materials and Methods

The articles included in this review were searched on PubMed using the queries summarized in Table 1. We searched for nine diseases and pathological conditions that were assumed to be related to nutrition. These were sarcopenia, frailty, heart failure, chronic kidney disease, diabetes, stroke, dementia, osteoporosis, and fracture. To prevent search omissions, the query “malnutrition or undernutrition or nutrition” was used.

The publication years ranged from 2015 to 2023 and articles published before 2015 were included only when no other similar reports existed. The date of the final electric search was 26 April 2023.

Titles and abstracts of the retrieved articles were screened to select potentially relevant articles. Full texts were also independently analyzed to determine whether they were eligible for inclusion. References in these articles were manually searched to identify further relevant articles. Since “aged 65 and older” was not included in the search query, articles targeting populations other than the elderly individuals were also included. These articles were not excluded because their results could have important implications for the nutritional management of each disease.

The articles were prioritized in the following order: systematic reviews, meta-analyses, randomized controlled trials (RCTs), prospective cohort studies, cross-sectional studies, and narrative reviews. Case reports, animal experiments, and articles written in languages other than English were excluded.

## 3. Results

### 3.1. The Number of Retrieved Articles

The number of articles retrieved by each query are summarized in Table 1.

### 3.2. The Analysis of Retrieved Articles

#### 3.2.1. Sarcopenia and Nutrition

In total, 4926 articles were retrieved, including 201 systematic reviews, 119 meta-analyses, 396 randomized controlled trials, 1179 cohort studies, and 56 guidelines. The number of articles increased with each year, as shown in Figure 2. Among the systematic reviews and meta-analyses, there were articles that explored community-dwelling elderly people (71 articles), cancer patients (48 articles), nutritional interventions and physical exercise (34 articles), inflammation or antioxidant (9 articles), renal disease (7 articles), liver cirrhosis (6 articles), and COVID-19 (6 articles).

We found 13 systematic reviews wherein the authors explored nutritional interventions in elderly individuals. Four systematic reviews showed the beneficial effects of leucine supplementation [42,43,44,45]. Three systematic reviews demonstrated the effect of β-hydroxy-β-methylbutyrate (HMB) on muscle mass and strength [46,47,48]. Scholars also reported the effects of cow-milk protein supplementation on muscle mass [49] and the effect of whey protein on physical performance [50]. In addition, three systematic reviews demonstrated the effectiveness of nutritional interventions [51,52,53]. However, the authors of another report found no evidence that protein or amino acid supplementation increased muscle mass or strength in predominantly healthy elderly people [54].

We found 24 systematic reviews with a pooled data analysis whereby the authors explored nutritional interventions and physical exercise. Fifteen systematic reviews reported the additional effects of nutritional interventions [55,56,57,58]. However, nine systematic reviews found that the additional effect of nutritional interventions were not significant or had limited significance [59,60,61,62]. In particular, the additional effects of nutritional interventions were not significant in patients with sarcopenic obesity [63,64]. An additive effect of resistance exercise and vitamin D3 supplementation on the increase in muscle strength was reported [65,66].

Nutritional interventions were performed in patients with several types of cancer or diseases, including pancreatic cancer [67], gastrointestinal cancer [68], liver cirrhosis [69], and CKD [70], to reduce sarcopenia and its complications.

For sarcopenic cirrhotic patients, an increase in protein and energy intake is recommended in the practical guidelines for liver disease of The European Society for Clinical Nutrition and Metabolism [71]. The same guideline also recommends an increase in protein intake in patients with active inflammatory bowel disease (IBD) and acute pancreatitis for obese patients who also have gastrointestinal and liver diseases to reduce sarcopenia risk [72].

#### 3.2.2. Frailty and Nutrition

In total, 3134 articles were retrieved, including 130 systematic reviews, 77 meta-analyses, 246 randomized controlled trials, 943 cohort studies, and 48 guidelines. The number of articles is also increased each year, as shown in Figure 2. 

Eight systematic reviews and meta-analyses explored nutritional interventions and physical exercise. Seven articles explored community-dwelling elderly people [53,73,74,75,76] and one article explored hospitalized patients [77]. The authors of six systematic reviews reported additional effects of nutritional interventions [53,73,74,77]. Two systematic reviews found that the additional effects of nutritional interventions were not significant or had limited significance [75,76].

Clinical practice guidelines of nutrition and physical activity have been proposed [78].

#### 3.2.3. Heart Failure and Nutrition

In total, 3609 articles were retrieved, including 110 systematic reviews, 104 meta-analyses, 240 randomized controlled trials, 1090 cohort studies, and 115 guidelines. Among the systematic reviews and meta-analyses, there were articles that explored outcomes and mortality (13 articles), nutritional interventions (7 articles), and etiology and epidemiology (3 articles).

The poor prognosis of heart failure patients associated with malnutrition was shown by using several nutritional assessment tools or diagnostic criteria; for example, Mini Nutritional Assessment (MNA) [79,80], Geriatric Nutritional Risk Index (GNRI) [81,82,83], Global Leadership Initiative on Malnutrition (GLIM) criteria [80], Controlling Nutritional Status (CONUT) [83,84]. A systematic review reported that nutritional interventions could potentially enhance outcomes in patients with malnutrition [85].

Coenzyme Q10 supplementation may reduce all-cause mortality [86,87]. Omega-3 fatty acids supplementation, represented by docosapentaenoic acid (DHA) and eicosapentaenoic acid (EPA) may be effective in preventing cardiovascular disease [88]. The Mediterranean diet may reduce incidence of heart failure [87,89].

#### 3.2.4. CKD and Nutrition

In total, 6080 articles were retrieved, including 163 systematic reviews, 138 meta-analyses, 425 randomized controlled trials, 1633 cohort studies and 185 guidelines.

Four systematic reviews and meta-analyses focused on the malnutrition of CKD patients [90,91,92,93]. Three systematic reviews and meta-analyses focused on sarcopenia and frailty in CKD patients [94,95,96]. The authors of three systematic reviews and meta-analyses suggested that oral nutritional supplements (ONS) containing protein may improve nutritional status of CKD patients [97,98,99].

Eight systematic reviews and meta-analyses discussed the effect of dietary protein restrictions on CKD progression and mortality. A reduction in the rate of the decline in renal function [100,101,102,103] and an improvement of nutritional status [104] were reported. However, three systematic reviews and meta-analyses found that the effect was limited or uncertain [105,106].

A higher risk of fall and fracture in CKD patients was reported [107]. Additionally, the association between mineral bone disorders (MBDs) and CKD was suggested [108,109]. The effect of vitamin D supplementation was analyzed. However, no beneficial effect on CKD–mineral bone disorders was observed [110,111,112], although an improvement in glycemic control, lipid profiles, and inflammation was noted [113].

Scholars have attempted to use dietary fiber and probiotics to treat dysbiosis in CKD patients, but the effects were limited [114,115].

#### 3.2.5. Diabetes and Nutrition

Contrary to the number of retrieved articles (46,363 articles, including 1605 systematic reviews and 1411 meta-analyses), as shown in Table 1, systematic reviews and meta-analyses which explored the nutritional status as well as sarcopenia and frailty in type 2 diabetes mellitus patients were rare. However, several narrative reviews and cross-sectional studies concerning poor nutritional status [116,117,118,119,120,121,122,123], as well as sarcopenia [121,123,124,125,126,127,128,129], and frailty [120] in diabetes mellitus patients were found.

Zinc (Zn) [130,131], vitamin E [132], folate [133], and L-arginine [134,135] supplementation and fiber-rich diets [136] may improve glycemic control and insulin resistance (IR) in type 2 diabetes mellitus patients. The Mediterranean diet may improve glycemic control [137] and reduce cardiovascular diseases risk [137,138]. The Nordic dietary pattern may also improve insulin resistance [139] and reduce obesity and cardiovascular diseases risk [140]. Evidence-based recommendations for the dietary management of diabetes mellitus patients have been proposed by The Diabetes and Nutrition Study Group of the European Association for the Study of Diabetes [141].

A correlation between poor nutritional status and foot ulcerations or a delay in wound healing was reported [142,143].

#### 3.2.6. Stroke and Nutrition

Among the systematic reviews and meta-analyses, 21 articles explored prevention of stroke, and 18 articles explored complication and outcomes.

Obesity is a risk factor for stroke [144]. However, several articles reported a better outcome in obese or overweight stroke patients, paradoxically [145].

High serum vitamin D levels were related to a reduction in stroke risk [146], and stroke recurrence [147]. A reduction in stroke risk due to the intake of vitamin B complex [148,149], vitamin E [150], fish [151,152], and a low-carbohydrate diet [153] was reported.

Post-stroke oropharyngeal dysphagia has been examined as one of the major complications associated with poor outcome [154,155,156]. The presence of malnutrition in stroke patients and its relationship with poor outcomes was reported [157,158,159]. However, the effect of nutritional interventions for stroke patients was not significant [160,161]. In cases where patients were undergoing enteral nutrition, improvement in their nutritional status, and the prevention of infection events and gastrointestinal complications by nutritional support combined with probiotics was reported [162,163]. 

The prevalence of stroke-related sarcopenia was also reported [164,165].

#### 3.2.7. Dementia and Nutrition

Among the systematic reviews and meta-analyses, there were articles that explored the risk and prevention of dementia (44 articles), nutritional interventions for elderly individuals with dementia (23 articles), and dysphagia and the application of enteral nutrition (17 articles).

Obesity was a risk factor for cognitive impairment and dementia [166,167], and underweight was also a risk factor for all-cause dementia [166]. The relationship between higher serum vitamin D levels and lower dementia and Alzheimer’s disease risk was reported [168,169,170]. Whether vitamin B complex prevents cognitive decline was still controversial [171,172]. An inverse association between fish consumption and dementia risk was observed by the authors of a meta-analysis of prospective studies [173]. Associations between adherence to the Mediterranean diet or a high consumption of fruits and vegetables and the reduction in mild cognitive impairment (MCI) and Alzheimer’s disease risk was reported [174,175].

The effect of nutritional interventions containing omega-3 fatty acid, represented by DHA and EPA, on cognitive function, blood amyloid-β-related biomarkers and inflammatory factors was controversial [176,177,178,179]. The effect of supplementing a single vitamin to enhance or maintain cognitive function was also still unclear [180]. Scholars had suggested that nutritional support combined with physical exercise can improve global cognitive function in elderly individuals with cognitive decline [181].

The energy expenditure of elderly individuals with dementia varies between individuals. Some individuals, particularly those who were community-dwelling exhibited relatively higher energy expenditure levels [182]. However, distinct eating disorder or dysphagia also occurred in elderly individuals with dementia. Malnutrition and unintended weight loss were frequently reported in patients with mild cognitive impairment, Alzheimer’s disease, and all-cause dementia [183,184,185,186]. Scholars had conducted several studies, including randomized controlled trial, on mealtime interventions to prevent eating disorders in individuals with dementia [183,184,185,186,187,188,189]. However, these studies were heterogeneous, and evidence regarding the increase in the dietary intake, nutritional status, activities of daily living (ADL), and quality of life of patients was still insufficient.

Enteral nutrition via a nasogastric tube or percutaneous endoscopic gastrostomy resulted in limited benefits in terms of the survival, behavioral and psychological symptoms, and quality of life of individuals with dementia [190]. Several decision guides on eating and drinking for people with severe dementia had been advocated [191].

#### 3.2.8. Osteoporosis and Nutrition

Among the systematic reviews and meta-analyses, eight articles where the authors explored osteoporosis prevalence in patients with various diseases were found. CKD–mineral bone disorders [108] and osteoporosis [109] in CKD patients were discussed. A high osteoporosis prevalence was reported in patients with chronic obstructive pulmonary disease (COPD) [192].

The association between low protein intake and decreased bone mineral density (BMD) or osteoporosis was reported [193,194]. A higher protein intake may have a protective effect on the bone mineral density of the lumbar spine compared with a lower protein intake [193].

The authors of many guidelines recommended vitamin D and calcium supplements to prevent osteoporosis [195,196,197]. The protective effect of vitamin D3 supplementation on the bone mineral density of the lumbar spine, femoral neck, and total hip was reported [198]. The authors suggested that calcium carbonate and vitamin D3 supplementation combined with nutritional interventions can enhance bone metabolism and bone mineral density of osteoporosis patients [199]. Vitamin K2 supplementation had a positive effect on the maintenance and improvement of bone mineral density in postmenopausal women [200].

#### 3.2.9. Fracture and Nutrition

Among the systematic reviews and meta-analyses, there were articles that explored the fracture risks (42 articles), malnutrition and nutritional interventions in hip fracture patients (10 articles), and the lowered risk of fractures achieved by supplementing vitamin D (6 articles).

The relationship between dietary patterns and fracture risk was discussed [201,202,203,204,205]. However, these studies, including systematic reviews, were heterogeneous. The beneficial effect of adhering to the Mediterranean diet regarding the incidence of hip fracture was reported [206,207]. The relationship between milk and dairy product consumption and hip fracture risk was discussed [208,209,210,211]. However, the reduction in risk or the protective effect caused by consuming milk or dairy products was still controversial. The protective effect of fracture by intake of vitamin A [212], vitamin C [213,214], and vitamin D [215,216,217] was reported. Vitamin D3 oral supplementations with or without calcium reduced hip and nonvertebral fractures in elderly individuals [218,219,220]. A positive association between saturated fatty acid (SFA) intake and hip fracture risk was reported [221].

Sarcopenia [222,223] and obesity [224,225] were risk factors for fractures independently; in addition, sarcopenic obesity was associated with an even higher risk of fractures compared with sarcopenia and obesity alone [226].

The increased risk of fracture in CKD [107] and diabetes mellitus [227] patients was reported. Both increased HbA1c levels and hypoglycemia may increase the risk of fracture in patients with diabetes mellitus [228].

A high prevalence of malnutrition was reported in hip fracture patients [229,230,231]. The beneficial effects of nutritional interventions, including optimization of nutritional intake and using oral nutritional supplements containing a high amount of protein in hip fracture patients were reported [232,233,234,235,236,237].

## 4. Discussion

In this review, we provide an overview of the literatures to determine whether nutrient deficiencies relate to the risk of diseases and whether nutrient intake can prevent the onset of diseases or enhance treatment outcomes (Figure 3). Malnutrition and nutrient deficiencies have been associated with an increased risk of CKD [90,91,92,93], stroke [157,158,159], osteoporosis [193,194], fractures [229,230,231], and dementia [183,184,185,186]. Nutritional interventions can reduce the risk and improve the prognosis of these diseases [97,98,99,193,232,233,234,235,236,237].

Although a consensus on nutritional management for sarcopenia and frailty has not been established, numerous studies have been conducted on this topic in recent years, as shown in Figure 2 [18,19,20,21,22,23,24,25,26,27,28,29,30,42,43,44,45,46,47,48,49,50,51,52,53,54,55,56,57,58,59,60,61,62,63,64,65,66,67,68,69,70,71,72,73,74,75,76,77,78,94,95,96,121,124,126,127,128,129,164,165,222,223,226]. The need for oral nutritional supplements to reduce sarcopenia [42,43,44,45,46,47,48,49,50,51,52,53,54,55,56,57,58,59,60,61,62,63,64,65,66,67,68,69,70,71,72] and frailty [53,73,74,75,76,77,78] is still controversial. According to The European Society for Clinical Nutrition and Metabolism guidelines on clinical nutrition and hydration in geriatrics [238], “Older persons with malnutrition or at risk of malnutrition with chronic conditions shall be offered oral nutritional supplements when dietary counseling and food fortification are not sufficient to increase dietary intake and reach nutritional goals”. This indicates the importance of the appropriate usage of oral nutritional supplements.

Protein supplementation may have potential benefits when it comes to malnutrition in CKD patients [97,98,99], in preventing osteoporosis [193], and treating hip fractures in the elderly [232,233,234,235,236,237]. Although a consensus has still not been reached, it is important to note that protein intake may need to be restricted in CKD patients to preserve renal function [100,101,102,103]. Considering that CKD is not uncommon in the elderly individuals, the protein intake of elderly individuals needs to be carefully evaluated. Leucine and HMB are effective in reducing sarcopenia [42,43,44,45,46,47,48,57]. HMB can be used without increasing the nitrogen load in CKD patients.

Preventing a vitamin D deficiency and supplementing it may have potential benefits with regard to reducing sarcopenia [65,66] and stroke risk [146,147], preventing decreased bone density [195,198,199], and lowering fracture risks [215,216,217,218,219,220]. Although the effectiveness of vitamin D supplementation at managing mineral bone disorders in CKD patients has not been demonstrated [110,111,112], they are relatively safe, and may be practically considered for their potential to improve glycemic control, lipid profiles, and inflammation [113]. Vitamin D is also being investigated for its potential association with lowering risk of developing dementia [168,169,170].

Evidence of malnutrition in patients with diabetes mellitus is limited [116,117,118,119,120,121,122,123], although scholars have suggested associations between diabetes mellitus and sarcopenia [121,123,124,125,126,127,128,129] and frailty [120]. Appropriate management procedure for malnutrition in diabetes mellitus patients has not still been established. Zn [130,131], vitamin E [132], folate [133], and L-arginine [134,135] supplementation and fiber-rich diets [136] have been suggested to potentially improve blood glucose control and insulin resistance. The reason why L-arginine improved glycemic control and insulin resistance is not clear in detail. L-arginine is a precursor for nitric oxide (NO) production by NO synthetase (NOS), and asymmetric dimethylarginine (ADMA) is a competitive inhibitor of NOS [239]. Elevated asymmetric dimethylarginine levels in diabetes mellitus patients correlate with the development of microangiopathies, including retinopathy and nephropathy [240]. L-arginine also stimulates insulin secretion from β-cells [241]. These properties may lead to improved glycemic control in diabetes mellitus patients. However, the evidence for this topic remains insufficient.

The Mediterranean diet has the potential in preventing heart failure [87,89], the impairment of glycemic control in diabetes mellitus patients [137,138,141], dementia [174,175], and fractures [206,207]. Omega-3 fatty acids, which is abundant in fish oil, may be beneficial in reducing heart failure [88] and dementia [177,178] risk. Additionally, the supplementation of vitamins may contribute to the prevention of various diseases, including vitamin E [132] and folate [133] for glycemic control and insulin resistance in diabetes mellitus patients; vitamin B complex [148] and vitamin E [150] for stroke risk; vitamin B complex, vitamin C, vitamin E and folate [172] for cognitive impairment; vitamin C [214] and vitamin K2 [200] for osteoporosis; and vitamin A [212] and vitamin C [213] for fractures. Obesity is a risk factor for all-cause mortality [242], diabetes mellitus [124,141], cardiovascular disease [243], ischemic stroke [144], cognitive impairment [166,167], fractures [224,225], and many other diseases [72]. Preventing obesity is fundamental to a healthy life. However, low BMI and decreasing lean body mass are also risks of mortality in elderly patients [145,244]. Thus, proper body weight management for elderly people and providing nutritional education for individuals and enlightening society as a whole is important.

The prevalence of malnutrition alone without any diseases is high in the elderly [245,246], and the presence of sarcopenia, frailty, and comorbidities may further increase their malnutrition risk. Therefore, screening and assessment for malnutrition is crucial. In the context of malnutrition and sarcopenia, nutritional assessment or diagnosis, represented by the Mini Nutritional Assessment [247,248] and GLIM [249], recommends the measurement of the skeletal muscle mass or the calf circumference [248,250,251].

Dietary fiber is an important material for short-chain fatty acids (SCFAs) production in the gut. SCFAs play an important role in gut barrier and microbiota maintenance, and they are associated with decreased inflammatory reactions [252]. Prebiotics and probiotics enhance the effect of dietary fiber. Dysbiosis, which is defined as the impairment of microbiota composition and gut barrier integrity, is associated with many chronic diseases, including diabetes mellitus [253] and CKD [254]. We expected to find evidence of the potential benefits that dietary fiber, prebiotics, and probiotics have for various health conditions, particularly by improving dysbiosis in CKD patients. However, although the effect has been shown in animal experimental models [255], evidence of this topic is still insufficient [114,115].

This review has several limitations. First, it is narrative and primarily based on previously published systematic reviews and meta-analyses. Moreover, the variety of diseases addressed in this review is limited. The influences of nutrients for patients with each type of cancer have also not been examined in detail. Additionally, exploratory research or small randomized controlled trials are not included.

## 5. Conclusions

Nutritional assessments and interventions for elderly patients, nutritional education for individuals, and the enlightenment of society should contribute to a reduction in sarcopenia, frailty and comorbidities in a super-aged society.

## Figures and Tables

**Figure 1 nutrients-15-02991-f001:**
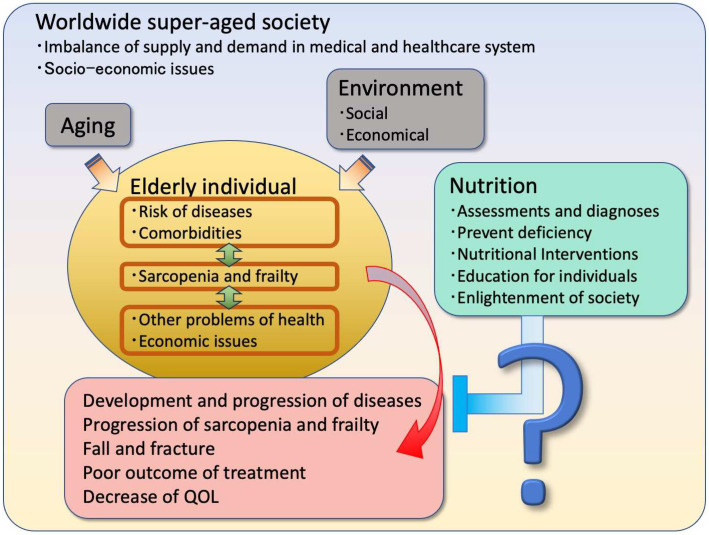
Our aim and questions addressed in this review. The arrows indicate the enhancement effects. Arrow with T-shaped head indicates an inhibitory effect.

**Figure 2 nutrients-15-02991-f002:**
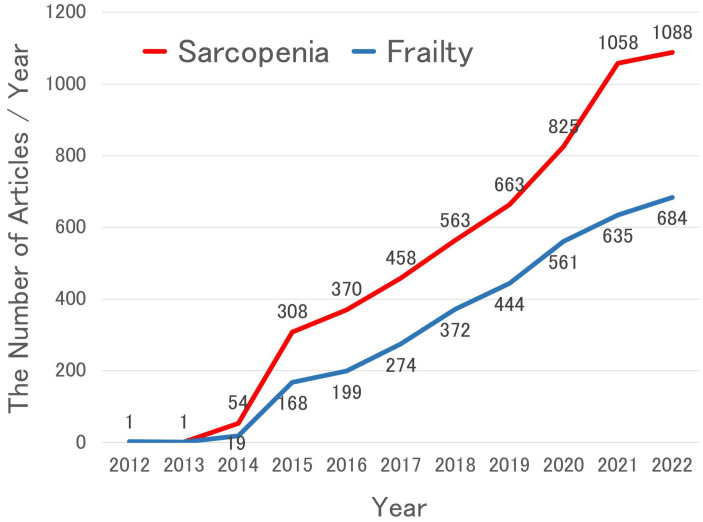
The number of retrieved articles concerning sarcopenia and frailty (per year).

**Figure 3 nutrients-15-02991-f003:**
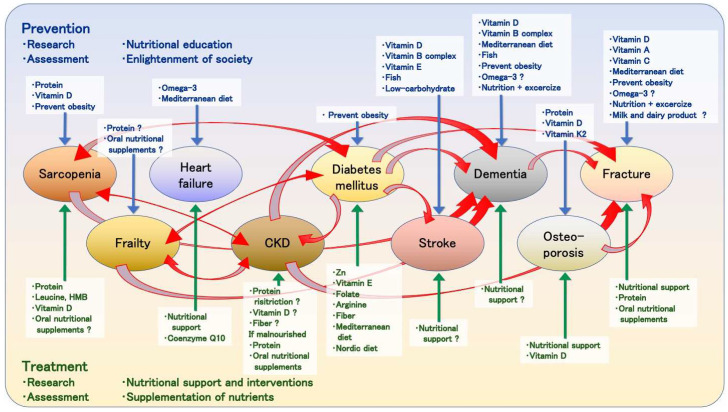
Relationship between diseases and nutrients. The upper part of the figure indicates the involvement of nutrients in prevention, and the lower part indicates their involvement in treatment. Nutrients written in blue letters with blue arrows indicate that the nutrients are involved in prevention of diseases. Nutrients written in green letters with green arrows indicate that the nutrients are involved in the treatment of the diseases. The red arrows indicate the relationship between diseases.

**Table 1 nutrients-15-02991-t001:** The number of articles retrieved by each query.

Query	Number of Articles
#1 sarcopenia and (malnutrition or undernutrition or nutrition) and (ENGLISH[LA] and 2015:2023[DP]	4926
#1 and (systematic review[PT] or meta-analysis[PT] or randomized controlled trial or cohort study or guideline)	1721
#1 and systematic review[PT]	201
#1 and meta-analysis[PT]	119
#1 and randomized controlled trial	396
#1 and cohort study	1179
#1 and Cochrane Database Syst Rev	0
#1 and guideline	56
#2 frailty and (malnutrition or undernutrition or nutrition) and (ENGLISH[LA] and 2015:2023[DP]	3134
#2 and (systematic review[PT] or meta-analysis[PT] or randomized controlled trial or cohort study or guideline)	1279
#2 and systematic review[PT]	130
#2 and meta-analysis[PT]	77
#2 and randomized controlled trial	246
#2 and cohort study	943
#2 and Cochrane Database Syst Rev	0
#2 and guideline	48
#3 heart failure and (malnutrition or undernutrition or nutrition) and (ENGLISH[LA] and 2015:2023[DP]	3609
#3 and (systematic review[PT] or meta-analysis[PT] or randomized controlled trial or cohort study or guideline)	1427
#3 and systematic review[PT]	110
#3 and meta-analysis[PT]	104
#3 and randomized controlled trial	240
#3 and cohort study	1090
#3 and Cochrane Database Syst Rev	7
#3 and guideline	115
#4 chronic kidney disease and (malnutrition or undernutrition or nutrition) and (ENGLISH[LA] and 2015:2023[DP]	6080
#4 and (systematic review[PT] or meta-analysis[PT] or randomized controlled trial or cohort study or guideline)	2221
#4 and systematic review[PT]	163
#4 and meta-analysis[PT]	138
#4 and randomized controlled trial	425
#4 and cohort study	1633
#4 and Cochrane Database Syst Rev	16
#4 and guideline	185
#5 diabetes and (malnutrition or undernutrition or nutrition) and (ENGLISH[LA] and 2015:2023[DP]	46,363
#5 and (systematic review[PT] or meta-analysis[PT] or randomized controlled trial or cohort study or guideline)	14,932
#5 and systematic review[PT]	1605
#5 and meta-analysis[PT]	1411
#5 and randomized controlled trial	4285
#5 and cohort study	9561
#5 and Cochrane Database Syst Rev	38
#5 and guideline	838
#6 stroke and (malnutrition or undernutrition or nutrition) and ENGLISH[LA] and 2015:2023[DP]	5356
#6 and (systematic review[PT] or meta-analysis[PT] or randomized controlled trial or cohort study or guideline)	2132
#6 and systematic review[PT]	265
#6 and meta-analysis[PT]	283
#6 and randomized controlled trial	428
#6 and cohort study	1807
#6 and Cochrane Database Syst Rev	18
#6 and guideline	145
#7 dementia and (malnutrition or undernutrition or nutrition) and ENGLISH[LA] and 2015:2023[DP]	4348
#7 and (systematic review[PT] or meta-analysis[PT] or randomized controlled trial or cohort study or guideline)	1396
#7 and systematic review[PT]	184
#7 and meta-analysis[PT]	149
#7 and randomized controlled trial	285
#7 and cohort study	975
#7 and Cochrane Database Syst Rev	13
#7 and guideline	48
#8 osteoporosis and (malnutrition or undernutrition or nutrition) and ENGLISH[LA] and 2015:2023[DP]	3802
#8 and (systematic review[PT] or meta-analysis[PT] or randomized controlled trial or cohort study or guideline)	1056
#8 and systematic review[PT]	118
#8 and meta-analysis[PT]	92
#8 and randomized controlled trial	223
#8 and cohort study	723
#8 and Cochrane Database Syst Rev	4
#8 and guideline	91
#9 fracture and (malnutrition or undernutrition or nutrition) and ENGLISH[LA] and 2015:2023[DP]	3616
#9 and (systematic review[PT] or meta-analysis[PT] or randomized controlled trial or cohort study or guideline)	1362
#9 and systematic review[PT]	154
#9 and meta-analysis[PT]	125
#9 and randomized controlled trial	232
#9 and cohort study	1021
#9 and Cochrane Database Syst Rev	8
#9 and guideline	89

## Data Availability

No new data were created.

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
