# Peer review of "Can Nutrition Contribute to a Reduction in Sarcopenia, Frailty, and Comorbidities in a Super-Aged Society?"

_nutrients, 2023, doi:10.3390/nu15132991_

Round 1

Reviewer 1 Report

The authors conducted a thorough literature search on the crucial topic of aging and the significance of nutritional intervention. I have the following concerns -

The structure of the abstract needs to be clearer and effectively communicate the authors' intended focus on nutritional supplementation.

Line 16  - These underlying diseases are associated. What do authors mean by underlying disease – sounds vague.

Line 17 – “Diabetes and chronic kidney disease are also associated with bone fragility” – No connection with the previous line.

Line 18 - Nutritional support. Please add a line stating the need for nutritional support. What is the reason for needing nutritional support?

The authors have done a good work of collecting information on aging populations and nutritional supplements, but the presentation of the information needs improvement.

Line 48-51: I am having difficulty comprehending the reason for the inclusion of dementia out of other diseases mentioned in lines 46 and 47 in the introduction. Furthermore, the explanation of dementia appears to be incomplete.

Figure 1: Figure 1 could be improved in terms of its contents and graphics. Also, I needed clarification on what the circles meant.

Methodology: Could you provide a definition or explanation of "super-aged society"? Also, please provide clarification on whether the authors specifically focused on individuals aged 65 and older. Additionally, explain if the authors only selected the publications in countries with a super-aged society.

N/A

Author Response

We would like to thank the reviewer for very insightful comments which have greatly helped us to improve the quality of our manuscript.

> Line 16  - These underlying diseases are associated. What do authors mean by underlying disease – sounds vague.

I am sorry that the explanation of underlying diseases was not clear. I changed the explanation as following, “Underlying diseases, such as diabetes, heart failure, chronic kidney disease and also dementia”.

> Line 17 – “Diabetes and chronic kidney disease are also associated with bone fragility” – No connection with the previous line.

I added the following sentence to clarify the relationship with the previous sentence. “Fractures in the elderly people are associated with bone fragility.”, and changed the sentence as following, “Diabetes and chronic kidney disease are associated with bone fragility.”.

> Line 18 - Nutritional support. Please add a line stating the need for nutritional support. What is the reason for needing nutritional support?

I strongly appreciate the reviewer's comment on this point. There are various cases in nutritional support. I changed the sentence as following, “Nutritional support for chronic disease patients and sarcopenic individuals with adequate energy and protein intake, vitamin D supplementation, blood-glucose-level management for individuals with diabetes, obesity prevention, nutritional education for healthy individuals, and the enlightenment of society could be crucial to solving the problems in super-aging societies.”.

> Line 48-51: I am having difficulty comprehending the reason for the inclusion of dementia out of other diseases mentioned in lines 46 and 47 in the introduction. Furthermore, the explanation of dementia appears to be incomplete.

I am sorry that the explanations of reasons for the inclusion of dementia out of other diseases were not clear. I have revised that section.

To make it easier to understand the relationship that as the super-aged society progresses, the number of elderly people with dementia increases, first sentence was changed as following “As aging population has progressed, the number of individuals with dementia is increasing [31]”.

I added the following sentence and references to show the relationship between dementia and other diseases, “CKD, HF, and DM with poor glycemic control are at risk of developing dementia.”

  1. Vishwanath, S.; Qaderi, V.; Steves, C.J.; Reid, C.M.; Hopper, I.; Ryan, J. Cognitive Decline and Risk of De-mentia in Individuals With Heart Failure: A Systematic Review and Meta-Analysis. J Card Fail 2022, 28, 1337–1348, doi:10.1016/j.cardfail.2021.12.014.
  2. Viggiano, D.; Wagner, C.A.; Martino, G.; Nedergaard, M.; Zoccali, C.; Unwin, R.; Capasso, G. Mechanisms of Cognitive Dysfunction in CKD. Nat Rev Nephrol 2020, 16, 452–469, doi:10.1038/s41581-020-0266-9.
  3. Aranda, M.P.; Kremer, I.N.; Hinton, L.; Zissimopoulos, J.; Whitmer, R.A.; Hummel, C.H.; Trejo, L.; Fabius, C. Impact of Dementia: Health Disparities, Population Trends, Care Interventions, and Economic Costs. J Am Geriatr Soc 2021, 69, 1774–1783, doi:10.1111/jgs.17345.

I changed the explanation of dementia as following, “Dementia is a complex of diseases, including Alzheimer's disease (AD) [32] and vascular dementia [33]. Typical symptom of dementia is loss of memory, language, prob-lem-solving and other thinking abilities.”

> Figure 1: Figure 1 could be improved in terms of its contents and graphics. Also, I needed clarification on what the circles meant.

I am sorry that our figure contains much content, and was complicated. The circles with trapezoids meant symbol of elderly populations, however, they were confusing. I removed them, and added a simple circle, which shows elderly individual.

I adjusted size of fonts and symbols, and position of arrow.

“Establishment of appropriate medical and healthcare system” has been changed to “Imbalance of supply and demand in medical and healthcare system”

> Methodology: Could you provide a definition or explanation of "super-aged society"? Also, please provide clarification on whether the authors specifically focused on individuals aged 65 and older. Additionally, explain if the authors only selected the publications in countries with a super-aged society.

I appreciate the reviewer's comment on these points. I added a sentence including definition of super-aged society in Introduction section as following, “The WHO defines a super-aged society as the proportion of the population aged 65 and over exceeding 21%”.

I am sorry that our concept in searching articles was not clear. We selected 9 diseases and pathological conditions that were assumed to be related to nutrition in elderly individuals. Since “aged 65 and older" was not included in the search query, articles not only targeting the elderly individuals were also included. However, these articles were not excluded because their results could have important implications for the nutritional management of each disease. The same sentences were added in Materials and Methods section.

Reviewer 2 Report

In this review Yoshida et al. provided an overview of the literature to determine if nutrient deficiencies could be linked to the risk of diseases. Moreover, they analyzed whether nutritional interventions could reduce the risk of disease in a super-aged society.

This is a well-done literature review and I have a few minor concerns to improve the paper:

1. I think the discussion is too long, it could be shortened to make the text easier to read.

2. The number of acronyms makes reading difficult, especially because some acronyms are used a few times throughout the text and could be removed without losing the focus of each section.

3. In Figure 3 would be easier to interpret if the authors wrote the text concerning the nutrients involved in prevention in one colour and those involved in treatment in a different colour.

4. Line 312 discussion, delete the extended name of CKD, it has already been mentioned in the introduction.

5. Please check typos errors.

Minor editing of English language required

Author Response

We would like to thank the reviewer for very insightful comments which have greatly helped us to improve the quality of our manuscript. 

  1. I think the discussion is too long, it could be shortened to make the text easier to read.

In accordance with the reviewer's comment, we have shortened the Discussion by 861 words (-141 words).

  1. The number of acronyms makes reading difficult, especially because some acronyms are used a few times throughout the text and could be removed without losing the focus of each section.

I would like to confirm if this comment was pointing out that many abbreviations were used. I removed the following abbreviations. Other common abbreviations have been used as the original manuscript. Please let me know if my understanding is different from what you intended.

COPD, chronic obstructive pulmonary disease; DNSG, The Diabetes and Nutrition Study Group; EASD, The European Association for the Study of Diabetes; LS, lumbar spine; PEG, percutaneous endoscopic gastrostomy; UEG, United European Gastroenterology; GPP, good practice points

  1. In Figure 3 would be easier to interpret if the authors wrote the text concerning the nutrients involved in prevention in one colour and those involved in treatment in a different colour.

I am sorry that our figure contains much content, and was complicated. I changed colour of the texts and the arrows. I added the following sentences in the Figure legend, “Nutrients written in blue letters with blue arrows indicate the involvement in the prevention of the disease. Nutrients written in green letters with green arrows indicate the involvement in the treatment of the disease”.

  1. Line 312 discussion, delete the extended name of CKD, it has already been mentioned in the introduction.

I removed the extended name of CKD in Line 312. I found the same errors in Line 51, 114, 227, and 980. I also fixed them.

  1. Please check typos errors.

I am sorry that the check of typos errors was incomplete. I checked again.